# Role of ADAR1 on Proliferation and Differentiation in Porcine Preadipocytes

**DOI:** 10.3390/ani14081201

**Published:** 2024-04-17

**Authors:** Menghuan Yang, Jun Jiang, Ruimin Ren, Ning Gao, Jun He, Yuebo Zhang

**Affiliations:** 1College of Animal Science and Technology, Hunan Agricultural University, Changsha 410128, China; 15637612639@stu.hunau.edu.cn (M.Y.); jiangjun1121@126.com (J.J.); ruimin.ren@hunau.edu.cn (R.R.); gaon@hunau.edu.cn (N.G.); 2Key Laboratory of Livestock and Poultry Resources (Pig) Evaluation and Utilization, Ministry of Agriculture and Rural Affairs, Changsha 410128, China

**Keywords:** ADAR1, preadipocyte, proliferation, differentiation, RNA-seq

## Abstract

**Simple Summary:**

Fat deposition is essential for the productivity of livestock farming and is also of considerable importance for maintaining human health. Previous studies have provided a clue to explore the involvement of adenosine deaminase acting on RNA 1 (ADAR1) in adipogenesis. In this study, we confirmed that ADAR1 enhances the proliferation and suppresses the differentiation and apoptosis of porcine preadipocytes through over-expression and knockdown approaches. We also identify the genes and pathways that ADAR1 may affect in the regulation of preadipocyte proliferation. The findings provide novel insights that shed light on the molecular mechanisms underlying lipid accumulation.

**Abstract:**

Recent research has identified ADAR1 as a participant in the regulation of lipid accumulation in mice. However, there are no reports on the roles of ADAR1 in proliferation, apoptosis and differentiation of porcine preadipocytes. In this study, we investigated the role of ADAR1 in differentiation, proliferation and apoptosis of porcine preadipocytes using CCK-8, EdU staining, cell cycle detection, RT-qPCR, Western blot, a triglyceride assay and Oil Red O staining. The over-expression of ADAR1 significantly promoted proliferation but inhibited the differentiation and apoptosis of porcine preadipocytes. The inhibition of ADAR1 had the opposite effect on the proliferation, differentiation and apoptosis of porcine preadipocytes with over-expressed ADAR1. Then, the regulation mechanisms of ADAR1 on preadipocyte proliferation were identified using RNA-seq, and 197 DEGs in response to ADAR1 knockdown were identified. The MAPK signaling pathway is significantly enriched, indicating its importance in mediating fat accumulation regulated by ADAR1. The study’s findings will aid in uncovering the mechanisms that regulate fat accumulation through ADAR1.

## 1. Introduction

Adipose tissue is an important place for energy storage in the animal’s body and is mainly composed of fat cells. Fat deposition is due to an increase in the number and size of fat cells. The increase in the number of adipocytes is due to embryonic stem cell proliferation and differentiation into adipocytes, and the increase in the size of differentiated adipocytes is caused by continuous lipid filling and hypertrophy [1]. The process of adipocyte proliferation and differentiation is a change in gene expression patterns.

Adenosine deaminase acting on RNA (ADAR) is a multi-gene family found in varying numbers among different species. The ADAR family comprises three principal members: ADAR1, ADAR2 and ADAR3. The proteins encoded by these genes are highly conserved, with each having a different number of double-stranded RNA-binding domains and a highly conserved C-terminal deaminase domain [2]. However, it is noteworthy that only ADAR1 and ADAR2 possess the catalytic activity required for RNA editing to occur.

ADAR1 is the most extensively researched RNA editing enzyme. Ota et al. [3] has revealed that ADAR1 can interact with dicer, thereby promoting miRNA maturation. This discovery highlights a novel function of ADAR1 in RNA processing, which is independent of its deaminase activity. These findings have significant implications for the regulation of various biological processes by ADAR1. Functional studies have shown that ADAR1 is involved in regulating the transformation of the smooth muscle cell phenotype and vascular remodeling [4], mesenchymal–epithelial transition [5], endothelial cell function [6], innate immune responses [7], neurotransmitter breakdown metabolism [8] and the development of cancers such as pancreatic ductal adenocarcinoma [9], thyroid cancer [10] and hepatocellular carcinoma [11]. These research findings highlight the importance of ADAR1 in cellular biology and disease progression. A recent study has also discovered that ADAR1 plays an important role in regulating fat deposition. Yu et al. [12] reported that ADAR1 has the potential to suppress the development of adipocytes in mice and mitigate diet-induced obesity. Our previous study has identified a large number of RNA editing events in pig adipose tissue. We found that genes with tissue-specific RNA editing are notably enriched in signaling pathways related to fat deposition and the analysis of RNA-seq data has shown that ADAR1 is highly expressed in adipose tissue [13]. It is thought that ADAR1 may play a crucial role in the development and function of adipose tissue. By investigating the role of ADAR1 in pig adipocytes, we may gain a better understanding of the molecular mechanisms underlying fat deposition and obesity-related metabolic diseases. 

In the context of the current research, we investigated the functional role of ADAR1 in proliferation and differentiation in porcine preadipocytes. Utilizing RNA-seq analysis, we identified the genes modulated by ADAR1 among preadipocyte proliferation. The study’s findings will aid in uncovering the mechanisms that regulate preadipocyte proliferation and differentiation.

## 2. Materials and Methods

### 2.1. Cell Culture

Under aseptic conditions, subcutaneous backfat tissues of Ningxiang piglets less than 7 days old were quickly passed through 75% anhydrous ethanol, and then rinsed in PBS containing 2% double antibiotics; visible blood vessels and muscles were removed. Adipose tissues were minced into a mushy state and placed in a 50 mL centrifuge tube, then digested with Type I collagenase (Merck, Shanghai, China) in a 37 °C water bath for 1.5 h, inverting every 10 min. After digestion, the reaction was stopped with an equal volume of a complete culture medium. A 100 μm cartridge was used to remove the undigested fractions, and then centrifuged at 800 rpm for 10 min to obtain the preadipocytes. The preadipocytes were then resuspended and seeded in the complete medium. The cells were cultured in a complete medium comprising 89% DMEM/F12 (Thermo Fisher Scientific, Grand Island, NE, USA), 10% Fetal Bovine Serum (FBS) (TransGen Biotech, Beijing, China) and 1% penicillin–streptomycin (Thermo Fisher Scientific, Grand Island, NE, USA) per 50 mL. To induce adipogenic differentiation, an adipogenic differentiation medium (per 50 mL: 2.25 mg 3-isobutyl-1-methylxanthine (Merck, Shanghai, China); 0.22 μmol Insulin (Beyotime Biotechnology, Shanghai, China); 0.001 μmol Dexamethasone (Merck, Shanghai, China)) was added when the cells reached confluency. After 2 d of treatment with the differentiation medium, the maintenance medium was replaced, and subsequently replaced every 2 d. The adipogenic differentiation of porcine preadipocytes spanned 8 d.

### 2.2. Cell Transfection

siRNA sequences targeting ADAR1 and Negative control (NC) were crafted and produced by RiboBio (Guangzhou, China). Overexpression plasmids (pcDNA3.1-ADAR1) and their empty vector control (pcDNA3.1) were constructed by GENE CREATE (Wuhan, China). The plasmids were isolated following the manufacturer’s guidelines (Omega Biotek, Norcross, GA, USA). Lipofectamine 2000 (Thermo Fisher Scientific, Grand Island, NE, USA) was used to transiently transfect pcDNA3.1-ADAR1, pcDNA3.1, siADAR1 and NC into cells.

### 2.3. Flow Cytometry

Porcine preadipocytes were seeded in six-well plates. When the confluence of cultured porcine preadipocytes reached 60–70%, ADAR1 overexpression plasmids or siRNA were transfected into porcine preadipocytes using Lipofectamine 2000. After 24 h of transfection, the cells were digested, collected and treated according to the instructions of the cell cycle and apoptosis detection kit (Meilunbio, Dalian, China). The cells were subjected to cell-cycle and apoptosis analysis using flow cytometry.

### 2.4. Cell Counting Kit-8 Assay

The cell proliferation rate was determined using the Cell Counting Kit-8 (CCK-8) (Meilunbio; Dalian, China) kit. Pig preadipocytes were plated at a density of 5000 cells per well in a 96-well format. Upon reaching approximately 40% confluence, the cells were transiently transfected with vectors designed for either enhanced ADAR1 expression or ADAR1 knockdown. The viability of the preadipocytes was monitored at intervals of 0, 24, 48 and 72 h post-transfection using the CCK-8 kit. To each well, 10% CCK-8 reagent was added. Subsequently, the mixture was maintained at 37 °C for 4 h. After the incubation period, we measured the optical density (OD) at 450 nm using a versatile microplate reader (Infinite M200PRO TECAN, Tecan, Switzerland).

### 2.5. EdU Staining

Porcine preadipocytes were seeded at 5000 cells per well in a 96-well plate. After reaching 50% confluence, they were transfected with overexpression plasmids and siRNA. At 36 h post-transfection, cells were stained using the EdU staining kit (BioScience, Shanghai, China) to detect cell proliferation. Inverted fluorescence microscope images were acquired using an inverted fluorescence microscope for subsequent analysis and count using ImageJ2.

### 2.6. Oil Red O Staining

For Oil Red O staining (Merck, Shanghai, China), the mature adipocytes were washed three times with PBS and fixed with 4% paraformaldehyde for 10 min. After discarding the paraformaldehyde, the cells were fixed again with fresh 4% paraformaldehyde for 1 h, then washed three times with PBS. Subsequently, 2 mL of 60% isopropanol was added for 2–5 min, and then the cells were allowed to dry completely on a clean bench. Finally, the cells were stained with 1% filtered Oil Red O working solution in the dark for 10 min. Cells were then washed multiple times with PBS until clear, and the lipid droplets in the cytoplasm of adipocytes were examined using an inverted fluorescence microscope (Carl Zeiss AG, Jena, Germany).

### 2.7. Triglyceride Determination

For triglyceride determination, porcine preadipocytes were digested with trypsin, then terminated with an equal volume of the complete culture medium and transferred to a centrifuge tube. The cell pellets were collected by centrifugation at 800 rpm for 5 min. Next, 200 µL of cell lysis buffer was added, the solution was mixed and rested for 10 min and the supernatant was collected by centrifugation for triglyceride determination. The supernatant was then collected by centrifugation after heating at 70 °C for 10 min to prepare a standard curve. Finally, triglycerides with an absorbance of 550 nm were measured using a multifunctional microplate reader (Infinite M200PRO TECAN).

### 2.8. RT-qPCR Analysis

RT-qPCR experiments were performed with the All-In-One 5× RT MasterMix (Applied Biological Materials Inc., Richmond, BC, Canada) following the protocol. The house-keeping gene is GADPH. The expression levels of the target genes were calculated with the 2^−∆∆Ct^ method.

### 2.9. Western Blot Detection

Proteins were prepared by using the Cell Lysis buffer (Tris-HCl) (Bioss, Beijing, China). The concentrations of proteins were measured with the Bicinchoninic Acid (BCA) assay method (Meilunbio, Dalian, China). Ten micrograms of proteins were separated by electrophoresis through SDS-polyacrylamide gels (YEASEN, Shanghai, China). The proteins obtained were then transferred to a polyvinylidene fluoride (PVDF) (Merck, Shanghai, China) membrane. The membrane was blocked with a rapid blocking solution for 2 h and then incubated with primary antibodies at 4 °C for 14 h. Subsequently, the PVDF membrane was washed and then hatched with a second antibody for 2 h. Finally, the protein bands were visualized with chemiluminescence reagents (Meilunbio, Dalian, China) and quantified using the ImageJ2 program. The CyclinD1 (CCND1) and ADAR1 antibodies were purchased from Aifang biological; Proliferating Cell Nuclear Antigen (PCNA), Bcl-2-associated X protein (Bax) and cysteine-dependent aspartate specific protease 3 (Caspase-3) antibodies were from Prteintech; B-cell leukemia/lymphoma-2 (Bcl2) was from Zenbio (Chengdu, China); β-actin antibody was from Cell Signaling Technology (Danvers, MA, USA).

### 2.10. RNA Sequencing and Data Processing

RNA-seq libraries were prepared by transfecting siADAR1 and NC sequences into preadipocytes 36 h after transfection. A total of six libraries were prepared, representing both the treatment and NC groups. The RNA was extracted using the Trizol reagent (Takara, Dalian, China). A NanoPhotometer spectrophotometer (NANODROP 2000) (Thermo Fisher Scientific, Grand Island, NE, USA) was used to assess the RNA quality. The RNA quality requires an RNA concentration > 200 ng/μL and an RNA Integrity Number (RIN) > 4.0. The cDNA libraries were constructed and sequenced after passing the quality test. Library preparation and sequencing were performed by the Novegene (Beijing, China) company on a NovaSeqX Plus PE150 sequence analyzer (Illumina) (Novegene, Beijing, China).

Differentially expressed genes (DEGs) were identified using DESeq2 (|log2FC| ≥ 1, padj < 0.05). The clusterProfiler 3.8.1 software was used to perform GO and KEGG enrichment analyses for DEGs. The GO terms and KEGG pathways with a *p*-value of 0.05 or less are considered statistically significant.

### 2.11. Statistical Analysis

All data were expressed as mean ± SD. Figures were produced using the GraphPad Prism 5 software. The Student’s *t*-test was used to determine differences between groups with *p*-values < 0.05 considered significant. “*n* = 3” represents three technical replicates.

## 3. Results

### 3.1. Efficiency of ADAR1 Interference and Overexpression

To study the impact of ADAR1 on porcine preadipocytes, cells were separately transfected with the overexpression vector pcDNA3.1-ADAR1 and the interference fragment siADAR1. After 36 h of culture, cells were collected to assess overexpression and interference efficiency. In the overexpression group, ADAR1 mRNA levels were increased over 90 times compared to the control group (Figure 1A), and protein levels also significantly increased (Figure 1B). After interference, mRNA levels decreased by more than 65% compared to NC, with siADAR1-3 showing the highest interference efficiency (Figure 1C), and protein levels were simultaneously reduced (Figure 1D). Therefore, siADAR1-3 was used in subsequent experiments.

### 3.2. ADAR1 Promotes Proliferation of Porcine Preadipocytes

After transfection for 24, 48 and 72 h with the ADAR1 overexpression plasmid, cell viability was significantly enhanced compared to pcDNA3.1 (Figure 2A), which indicates that ADAR1 promoted cell proliferation. The EdU assay showed that increased ADAR1 expression resulted in an evident rise in the quantity of proliferating cells (Figure 2B). Flow cytometry results revealed a significantly decreased number of cells in G1 phase, and a significant increase in G2 and S phase (Figure 2C). In addition, RT-qPCR results showed significant upregulations of the cell proliferation marker genes CyclinE1 (CCNE1), CCND1, c-Myc proto-oncogene, bHLH transcription factor (CMYC), Bone morphogenetic protein 4 (BMP4) and Cyclin-dependent Kinase 4 (CDK4) (Figure 2D). WB results showed a significant increase in CCND1 and PCNA in response to ADAR1 overexpression (Figure 2E). These results indicate that the overexpression of ADAR1 promotes cell proliferation.

Following ADAR1 knockdown, cell viability significantly decreased compared to the control group after 24, 48 and 72 h of treatment, indicating that ADAR1 knockdown inhibited adipocyte proliferation (Figure 3A). EdU results showed a significant decrease in the number of proliferating cells (Figure 3B). Flow cytometry results revealed a significant increase in G1 phase cells and a significant decrease in G2 cells, with a downward trend in S phase cells (Figure 3C). RT-qPCR results showed a significant decrease in cell proliferation marker genes CCNE1, CCND1, CMYC, BMP4, CDK4 and PCNA (Figure 3D). WB results showed a significant decrease in CCND1 and PCNA protein levels (Figure 3E). These results suggest that ADAR1 knockdown inhibits cell proliferation.

### 3.3. ADAR1 Inhibits Apoptosis of Porcine Preadipocytes

To explore the effect of ADAR1 on apoptosis in porcine preadipocytes, we utilized flow cytometry, RT-qPCR and Western blotting to detect cell apoptosis. Flow cytometry apoptosis detection results showed that the proportion of early and late apoptotic cells was significantly lower in the ADAR1 overexpression group than that in the control group (Figure 4A). RT-qPCR and WB results showed that after overexpressing ADAR1, anti-apoptotic gene Bcl2 was significantly increased, while pro-apoptotic gene Caspase-3 and Bax were significantly decreased in mRNA/protein levels (Figure 4B,C). These results suggest that overexpression of ADAR1 inhibits cell apoptosis.

Next, we studied the effect of ADAR1 knockdown on cell apoptosis. Apoptosis detection by flow cytometry revealed a significant increase in the proportion of early and late apoptosis cells (Figure 5A). RT-qPCR and WB analyses revealed a significant reduction in the mRNA level of the anti-apoptotic gene Bcl2, as well as a reduction in protein levels, although the latter was not statistically significant. Conversely, both the protein and mRNA levels of the proapoptotic genes Caspase-3 and Bax were significantly increased (Figure 5B,C). Overall, these results suggest that ADAR1 knockdown promotes cell apoptosis.

### 3.4. ADAR1 Inhibits Differentiation of Porcine Preadipocytes

To investigate the impact of ADAR1 on the adipogenic differentiation of porcine preadipocytes, cells were engineered to overexpress ADAR1 for 24 h and then induced to differentiate for 8 d. Subsequently, they were stained with Oil Red O and assessed for triglyceride content (Figure 6A,B). Results showed that overexpression of ADAR1 inhibited adipogenic differentiation and triglyceride accumulation. RT-qPCR results indicated a significant decrease in the levels of the Peroxisome proliferator-activated receptor γ (PPAR-γ), CCAAT/enhancer binding protein α (C/EBPα), fatty acid synthase (FASN) and fatty acid binding protein 4 (FABP4) (Figure 6C). These results suggest that overexpression of ADAR1 inhibits the differentiation of porcine preadipocytes.

Then, we explored the effect of ADAR1 knockdown on adipocyte differentiation. Porcine preadipocytes were transfected and induced to differentiate for 8 d after 24 h of ADAR1 interference treatment. The Oil Red O staining and triglyceride assay demonstrated that ADAR1 knockdown promoted adipogenic differentiation and triglyceride accumulation in porcine preadipocytes (Figure 7A,B). RT-qPCR indicated that the expression levels of PPAR-γ, C/EBPα, FASN and FABP4 significantly increased (Figure 7C). The above results suggest that ADAR1 knockdown promotes the differentiation of porcine preadipocytes.

### 3.5. Analysis of Differentially Expressed Genes (DEGs)

To assess the impact of ADAR1 on the transcriptomic expression profile during proliferation, we performed RNA-seq analysis of pig preadipocytes transfecting with siADAR1 and NC sequences for 36 h. We first checked the knockdown effect of ADAR1 in the RNA-seq data and found that ADAR1 expression was effectively inhibited in both the NC and treatment groups, consistent with the qPCR results (Figure 8A). There were 197 DEGs, including 61 upregulated and 136 downregulated genes (Appendix A). We randomly selected 3 genes for validation using RT-qPCR, and consistent results were obtained (Figure 8B), which indicate the reliability of our RNA-seq data.

To investigate the functional roles of the DEGs, we performed a GO enrichment analysis on the 197 DEGs (Appendix A). These DEGs were classified into three categories, including 2351 terms for Biological Process (BP), 219 terms for Cellular Component (CC), and 358 terms for Molecular Function (MF). The top 20 GO terms were found to be statistically significant (*p* < 0.01). Overall, there were no terms significantly enriched in the Cellular Component (CC) category. Additionally, one term significantly enriched in the Molecular Function (MF) category is “lyase activity”. However, a total of 19 Biological Process (BP) terms were found to be significantly enriched, several of which are known to be involved in “tissue remodeling”, “regulation of growth”, “cellular modified amino acid catabolic process” and “cellular component maintenance” (Figure 8C, Appendix A). Following this, the analysis showed that the DEGs were significantly enriched in the “Cytokine-cytokine receptor interaction”, “MAPK signaling pathway”, “TNF signaling pathway”, “cAMP signaling pathway” and “IL-17 signaling pathway” (*p* < 0.05) (Figure 8D). Therefore, ADAR1 might regulate the proliferation of adipocytes through these key pathways.

We then identified 28 differentially expressed genes (DEGs) associated with cell proliferation by GO term categorisation. Among these DEGs, there were 10 DEGs with upregulation and 18 DEGs with down-regulation, including T-box 2 (TBX2), Fibroblast growth factor 12 (FGF12), GLI family zinc finger 1 (GLI1), Leukemia inhibitory factor (LIF) and Mitogen-activated protein kinase kinase kinase 7 (MAP3K7) (Appendix A).

## 4. Discussion

ADAR1 plays important roles in development and disease, including an increasing connection to cancer progression [14]. Recent research has revealed that ADAR1 plays a significant regulatory role in fat deposition. Yu et al. [12] found that ADAR1 can inhibit the adipogenic differentiation of mouse fibroblasts and obesity induced by a high-fat diet. However, the role of ADAR1 in pig adipocyte proliferation, differentiation and apoptosis functions remains unknown. Therefore, studying the function of ADAR1 during preadipocyte differentiation could have a significant impact on our understanding of obesity in mammals and on the meat quality in domestic animals.

In our research, we manipulated the expression levels of ADAR1 in porcine preadipocytes using the pcDNA3.1 vector for overexpression and siRNA for downregulation. This is the first report on the role of ADAR1 in increasing the growth rate of these early adipocytes. Similar observations regarding the influence of ADAR1 on cell proliferation have been documented. For example, Zhang et al. [15] showed that increased ADAR1 levels enhanced the progression of pulmonary hypertension (PH) and stimulated the growth of pulmonary artery smooth muscle cells (PASMCs). ADAR1 p110 subtype knockdown inhibits glioma cell proliferation [16]. Xiao et al. [17] also found that reducing the expression of ADAR1 was associated with a reduced rate of leukaemia cell proliferation outside a living organism in vitro. The distribution of preadipocytes in the different phases of the cell cycle is an indicator of their proliferation rate. An increase in the cell number of the S phase indicates a corresponding increase in cell proliferation activity [18]. Our data showed that ADAR1 overexpression significantly increased the number of S phase adipocytes. Song et al. [19] found that ADAR1 enhanced the percentage of cells in the S phase in HEK 293 cells. The influence of ADAR1 on the expansion of preadipocytes was further indicated through the expression levels of genes that act as cell proliferation markers. Overexpression of ADAR1 significantly upregulated mRNA and protein levels of proliferation-related genes such as PCNA and CCND1. The opposite is true for interfering with ADAR1. PCNA, a marker gene of proliferation, participates in DNA biosynthesis and regulates the cell cycle [20,21]. The CCND1 protein belongs to a highly conserved cyclin family that regulates CDK kinases [22].

The apoptosis pathways in eukaryotic cells include endogenous, exogenous and endoplasmic reticulum stress pathways, with the Caspase protein family playing a crucial role in this process. The results of this study showed that knockdown of ADAR1 significantly promoted the protein expression of caspase-3, suggesting that ADAR1 may inhibit cellular lipid deposition by promoting apoptosis of porcine preadipocytes. Previous studies in the DU145 and PC3 cell lines have shown that silencing ADAR1 reduces growth rate and induces cell death [23]. ADAR1 silencing induced HUVEC apoptosis [24]. However, the loss of ADAR1 in human iPS cells promotes caspase-3 mediated apoptotic cell death but did not change the proliferation rate [25]. These results indicate that ADAR1 has cell type differences in regulating apoptosis. In this study, the mRNA expression level of the apoptosis inhibitor Bcl2 was significantly decreased in the ADAR1 interference group, suggesting that ADAR1 may promote apoptosis by inhibiting the expression of Bcl2, thereby inhibiting intracellular lipid deposition, but whether ADAR1 directly targets or regulates Bcl2 through indirect pathways still needs further investigation. However, there are few studies on the relationship between ADAR1 and Bcl2, and the relationship between ADAR1 regulating Bcl2 expression and then participating in the regulation of intracellular lipid deposition processes needs to be studied.

Preadipocytes can differentiate into mature fat cells that accumulate lipid droplets under the influence of inducers. Lipid droplets are a significant morphological characteristic of mature fat cells and gradually accumulate during the process of adipocyte differentiation [26]. The finding revealed that ADAR1 suppressed the differentiation of porcine preadipocytes, which are known to be associated with the adipogenic differentiation process. FASN is an enzyme that is crucial in the de novo synthesis of fatty acids. Studies on mouse adipocytes have shown that upregulation of FASN expression can significantly increase the accumulation of intracellular triglycerides [27]. Both PPAR-γ and C/EBPα are essential for fat production, whether in vitro or in vivo [28]. PPAR-γ regulates the terminal differentiation of fat cells, and C/EBPα works in cooperation with PPAR-γ to activate gene expression transcription in mature fat cells [29]. C/EBPs are composed of six members and are key factors in regulating cell differentiation. Within the C/EBP family, C/EBPβ is expressed promptly after induction of adipogenesis by the adipogenesis mixture (MDI). Upon phosphorylation by the mitogen-activated protein kinase (MAPK), C/EBPβ gains the ability to bind DNA, subsequently activating the transcription of PPAR-γ and C/EBPα, both of which are essential transcription factors in the process of adipogenesis. In light of the above studies and our findings, it is inferred that ADAR1 inhibits adipogenic differentiation of porcine preadipocytes.

In the study, high-throughput sequencing technology was used to sequence the mRNA of two groups of porcine preadipocytes: one with ADAR1 inhibition and the other with normal expression. Analysis of data from transfected siRNA and NC cells revealed significant enrichment (*p* < 0.05) in several well-known adipose accumulation pathways, including the MAPK, cAMP, TNF and IL-17 signalling pathways. Besides to reducing inflammation, AMPK plays a key role in controlling the formation of fat cells (adipogenesis) in adipose tissue [30]. The MAPK signalling pathway regulates adipocyte differentiation and has been shown to downregulate the expression of C/EBPα and PPAR-γ, thereby regulating adipocyte differentiation [31,32]. IL-17 is a cytokine that plays a regulatory role in the formation of adipocytes and the growth of adipose tissue in the body [33]. In the GO analysis, there were many DEGs involved in cell proliferation, including TBX2, FGF12, GLI1, LIF and MAP3K7. TXB2 belongs to a group of eicosanoids called prostanoids, which is produced by cyclooxygenase and have been linked to inflammation and insulin resistance in the context of obesity [34,35,36]. Overexpression of FGF12 in primary human aortic vascular smooth muscle cells (HASMC) led to the development of a quiescent and contractile phenotype, which was achieved by activating the p38 MAPK pathway and inhibiting cell proliferation via the p53 pathway. In Contrast, when FGF12 was knocked down, cell proliferation was augmented and phenotype changes were inhibited [37]. Studies have shown that MAP3K7 is an important molecule in the MAPK signalling pathway, which activates downstream MAPK molecules, and thus participates in the regulation of cell growth and proliferation [38]. This suggests that the DEGs we screened are related to ADAR1 adipocyte development. The results provide insights into the relationship between the MAPK signalling cascade and the proliferation of preadipocytes.

## 5. Conclusions

In this study, we confirmed that ADAR1 enhanced the proliferation of porcine preadipocytes, and inhibited differentiation and apoptosis. Furthermore, we identified various genes involved in the regulation of ADAR1 in adipogenesis, including 28 known proliferation-related genes. ADAR1 might promote cell proliferation through the MAPK signalling pathway. This study provides the foundations for further revealing the mechanism of ADAR1 in regulating fat deposition.

## Figures and Tables

**Figure 1 animals-14-01201-f001:**
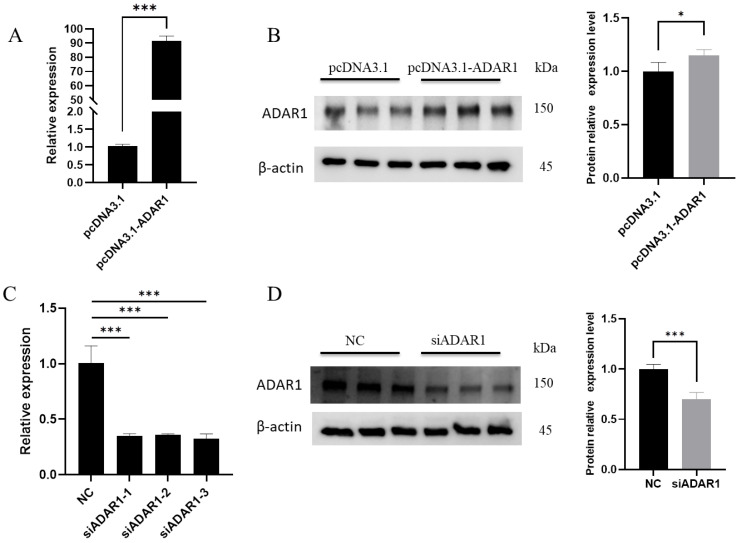
Detection of ADAR1 overexpression and knockdown efficiency. The efficiency of ADAR1 overexpression at mRNA (**A**) and protein (**B**) levels. The efficiency of ADAR1 knockdown at mRNA (**C**) and protein (**D**) levels. Results are shown as mean ± SD (*n* = 3). * *p* < 0.05, *** *p* < 0.005.

**Figure 2 animals-14-01201-f002:**
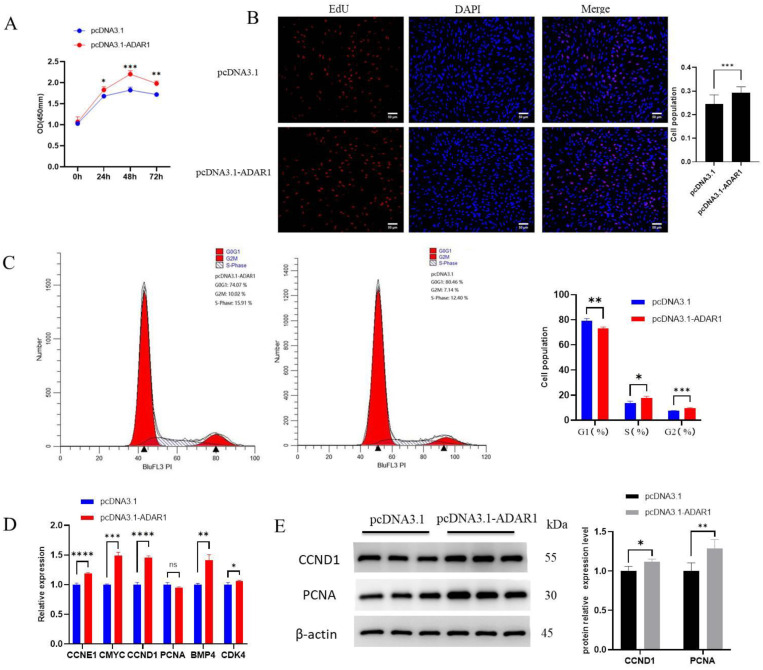
Overexpression of ADAR1 promotes preadipocyte proliferation. (**A**) CCK-8 cell viability assay. The absorbance value (OD) was measured at 450 nm. (**B**) EdU incorporation assay was used to determine the proliferation activity. (**C**) Flow cytometry analysis for cell cycle detection. (**D**) Relative mRNA expression of CCNE1, CMYC, CCND1, PCNA, BMP4 and CDK4 genes. (**E**) Protein expression analysis of CCND1 and PCNA with β-actin as the house-keeping protein. Results are shown as mean ± SD (*n* = 3). ^ns^
*p* > 0.05, * *p* < 0.05, ** *p* < 0.01, *** *p* < 0.005, **** *p* < 0.001.

**Figure 3 animals-14-01201-f003:**
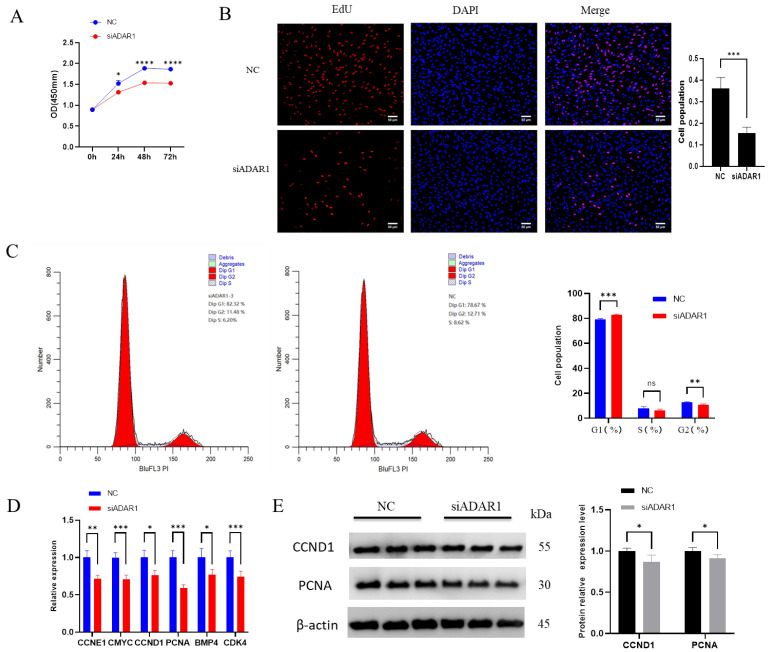
Interference with ADAR1 inhibits preadipocyte proliferation. (**A**) CCK-8 cell viability assay. The absorbance value (OD) was measured at 450 nm. (**B**) EdU incorporation assay was used to determine the proliferation activity. (**C**) Flow cytometry analysis for cell cycle detection. (**D**) Relative mRNA expression of CCNE1, CMYC, CCND1, PCNA, BMP4 and CDK4 genes. (**E**) Protein expression analysis of CCND1 and PCNA genes with β-actin as the house-keeping protein. Results are shown as mean ± SD (*n* = 3). ^ns^
*p* > 0.05, * *p* < 0.05, ** *p* < 0.01, *** *p* < 0.005, **** *p* < 0.001.

**Figure 4 animals-14-01201-f004:**
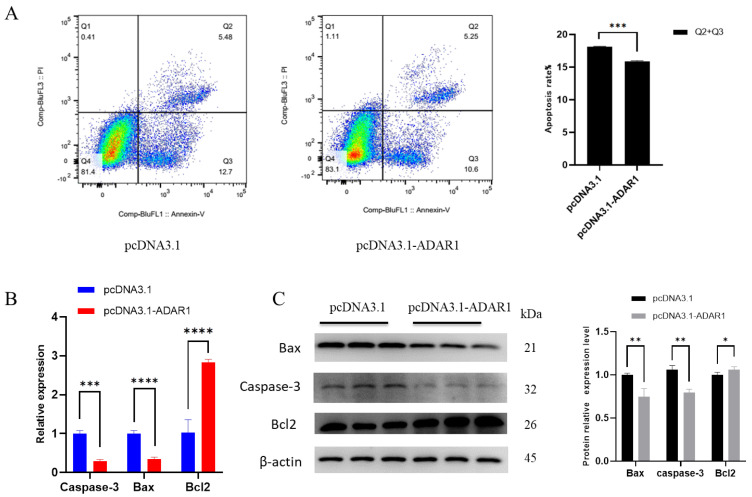
Overexpression of ADAR1 inhibits preadipocyte apoptosis. (**A**) Cell apoptosis rates detected by flow cytometry. (**B**) Relative mRNA expression of Caspase−3 and Bcl2 genes. (**C**) Protein expression analysis of Caspase−3 and Bax genes by Western blotting with β−actin as the house−keeping protein. Results are shown as mean ± SD (*n* = 3). * *p* < 0.05, ** *p* < 0.01, *** *p* < 0.005, **** *p* < 0.001.

**Figure 5 animals-14-01201-f005:**
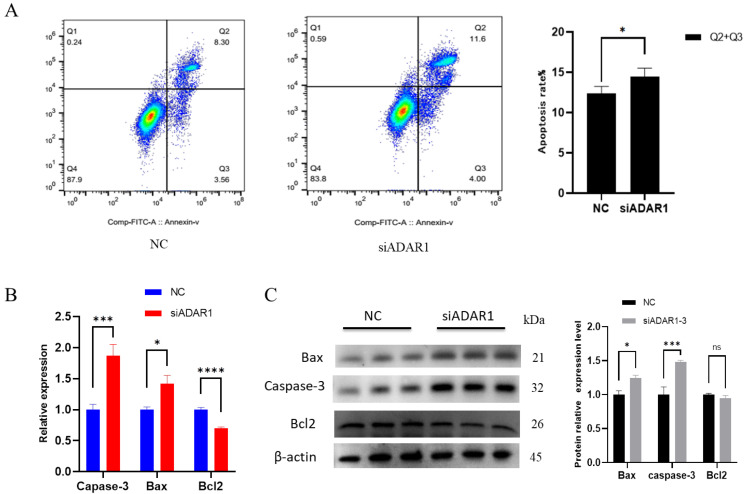
ADAR1 knockdown promotes preadipocyte apoptosis. (**A**) Cell apoptosis rates detected by flow cytometry. (**B**) Relative mRNA expression of Caspase-3 and Bcl2 genes. (**C**) Protein expression analysis of Caspase-3 and Bax genes with β-actin as the house-keeping protein. Results are shown as mean ± SD (*n* = 3). ^ns^
*p* < 0.05, * *p* < 0.05, *** *p* < 0.005, **** *p* < 0.001.

**Figure 6 animals-14-01201-f006:**
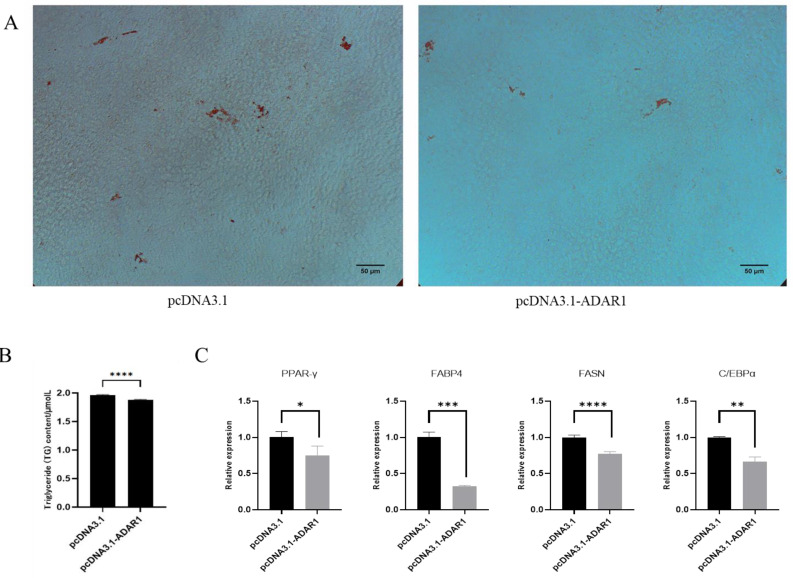
ADAR1 overexpression inhibits preadipocyte differentiation. (**A**) Oil Red O staining at 8d of porcine preadipocyte differentiation. (**B**) Detection of triglyceride content. (**C**) Relative mRNA expression of PPAR-γ, C/EBPα, FABP4 and FASN genes. Results are shown as mean ± SD (*n* = 3). * *p* < 0.05, ** *p* < 0.01, *** *p* < 0.005, **** *p* < 0.001.

**Figure 7 animals-14-01201-f007:**
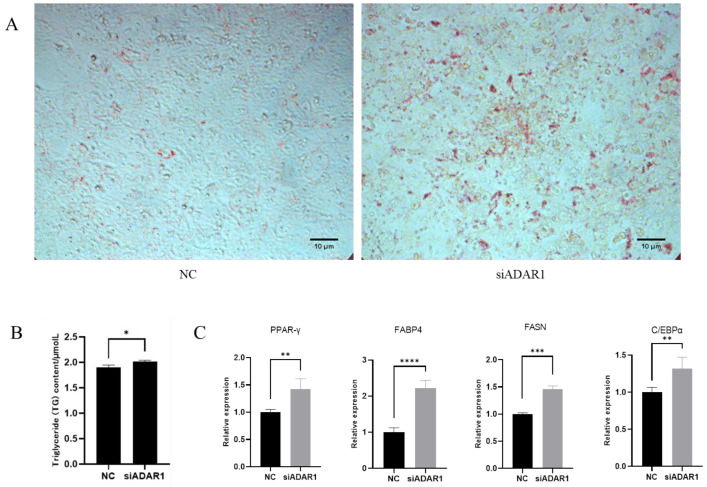
ADAR1 knockdown promotes preadipocyte differentiation. (**A**) Oil Red O staining at 8 d of porcine preadipocyte differentiation. (**B**) Detection of triglyceride content. (**C**) Relative mRNA expression of PPAR-γ, C/EBPα, FABP4 and FASN genes. Results are shown as mean ± SD (*n* = 3). * *p* < 0.05, ** *p* < 0.01, *** *p* < 0.005, **** *p* < 0.001.

**Figure 8 animals-14-01201-f008:**
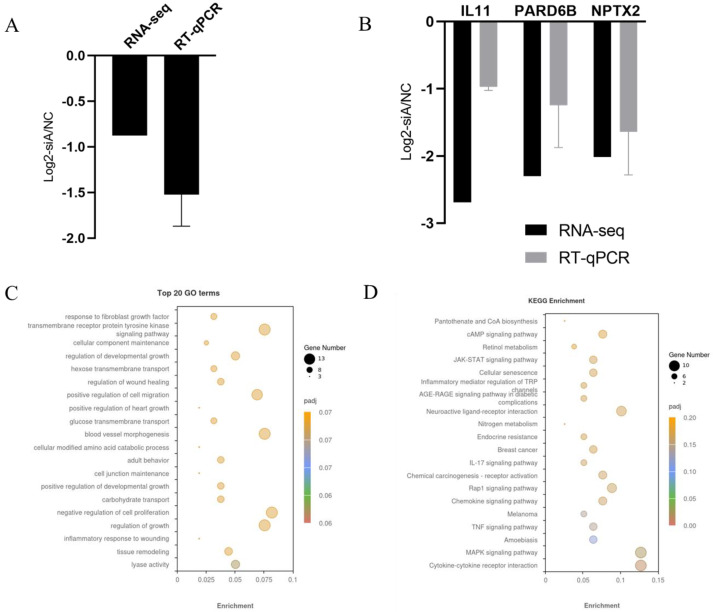
Overview of RNA−seq data. (**A**) ADAR1 is effectively downregulated by siRNA during the proliferation of preadipocytes. (**B**) Validation of RNA−seq data by RT−qPCR. (**C**) Top 20 GO terms enriched by DEGs. (**D**) KEGG pathways significantly enriched by DEGs.

## Data Availability

The raw data have been submitted to the BIG Submission (accession number PRJCA023839).

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
