# Peer review of "Role of ADAR1 on Proliferation and Differentiation in Porcine Preadipocytes"

_animals, 2024, doi:10.3390/ani14081201_

Round 1

Reviewer 1 Report

Comments and Suggestions for Authors

Dear Authors,

The presented manuscript identified ADAR1 function for proliferation and differentiation in porcine preadipocytes by using multiple methods: ADAR1 knockdown, ADAR1 overexpression, estimation of proliferation and differentiation capacity, identification of multiple proteins using WB, and estimation of triglyceride contents. The main concern is that in the study, only one pig was used, so we do not have the biological repeats but only technical repeats. However, cell culture studies often use only one biological replicate, so it is not discriminating.

In the materials and methods, it is given that the whole experiment was performed using one 7-year-old piglet, but it is not given what breed was used. Does the pig belong to fatty or lean pigs? Moreover, it should be given how many technical replicates each experiment was performed. Only during RNA-seq library preparation was it given that 6 libraries were prepared for treatments (siADAR1) and the NC group. Why was the whole transcriptome not measured after ADAR1 overexpression? but only genes associated with proliferation or differentiation were estimated; throughout the whole transcriptome analysis, we would have the full picture of the changes after ADAR1 overexpression. I know that the methods used proved the influence on proliferation or differentiation. Why did the authors decide to do transcriptome analysis after ADAR1 knockdown? This approach is not consistent. 

Which RNA quality was acceptable?

cDNA libraries were tested for their quality?

The measurements of apoptosis ADAR1 subsection (Inhibits Apoptosis of Porcine Preadipocytes). At the expression level, proapoptotic caspase3  and antiapoptotic bcl2 were measured, but at the protein level, only proapoptotic BAX and caspase 3 were measured. Why were not all genes measured at all levels?

Figure 4 What does mean n = 3, three technical replicates?, please clear

Figure 5 shows BAX, caspase3 and bcl2 expression levels and at the protein level, only BAX and caspase3; the question as above.

Figure 6A is hardly legible/visible. n = 3 Is it technical replicates collected from different plates or just technical replicates during qPCR?

 Figure 7 has a probable wrong heading. Should it be differentiation, not apoptosis?

Reviewer 2 Report

Comments and Suggestions for Authors

This study aimed to elucidate roles of ADAR1 in porcine adipogenesis in vitro. The authors addressed well to demonstrate the functions in the preadipocyte prolifelation, differentiation, and apoptosis. I reccomend this article for publication in Animals, after minor revision of the manuscript according to the points to be revised as listed below: 

 Throughout the text, all the gene names should be spelled out when they first appeared. 

In Materials and Methods, the authors should provide information company, the location, and the country for all the reagents and experimental tools you described here. 

line 71 and 78: Sell out FBS and NC.

line 159: what did you set at 0.05? p-value?

Figure 1 panel C: pair of difference should be indicated separately for each pair to be compared, because you mention that the difference was statistically tested by t-test. 

Figure 2 panel B: All the images are poor in quality, too dark to understand the positive staining. Change the images. Add scale bar in each image.

Figure 3 panel B? (no indication): All the images are poor in quality, too dark to understand the positive staining. Change the images.

Figure 6A: too small number of cells to evaluate the effect of ADAR1 overexpression  in these panels. Use alternative images with higher quality. I do not understand the significant difference. How many cells did you count for each field? This largely affects reliance of the data. 

Reviewer 3 Report

Comments and Suggestions for Authors

Yang et al. reported the role of ADAR1 on the proliferation and differentiation in porcine adipocytes by employing molecular biology techniques such as overexpressing and silencing, flowcytometry, and global RNA-sequencing. Data are compact although with a very small repetitions, n=3.

I would like to address several points following:

1. Authors should define the abbreviation in the first appearance of the terms. Many abbreviations were not defined properly, which were found in abstract. More importantly, ADAR1 was not defined either in abstract or indtroduction.

2. I would suggest to change the title from Effect of ADAR1.... to Role of ADAR1......

3. Introduction was not sufficient and required a significant improvement. The background that underlines the reason why authors selected ADAR1 was vague. In line 42-45, authors briefly wrote about their findings [reference 4], where authors identified a large number of RNA editing events in pig adipose tissue, was ADAR1 found in this finding as well?

4. In the method part, authors mentioned that the fat tissue was obtained from piglets' subcutaneous tissue, please clarify which part of subcutaneous? Abdominal or other part.

5. As mentioned above, please define the abbreviation. In the method part, NC (line 78) needs to be defined properly.

6. Please put molecular weight of the protein target in the immunoblotting data. I would suggest to put B-ACTIN below the protein targets.

7. In the methods, authors wrote that the cell counting was performed at 0, 14, 48, and 72 hours after transfection. In the results, which part/figure correlate with this? The cell population shown in Figure 2B and 3B only showed one time point, which required a clarification.

8. In figure 3, authors need to add panel "B" because it is missing.

9. In figure 4, authors showed immunoblotting for Bax and caspase-3, however no protein data for Bcl2. Adding Bcl2 protein data is advisable.

10. In Figure 6B and 7B, authors showed data for triglyceride, the difference is very small between NC and overexpressed ADAR1 or siADAR1, although it is significant. The change in FASN (fatty acid synthase) expression is quite large between NC and overexpressed ADAR1 or siADAR1 (Figure 6C and 7C). Please explain more detail about this data.

11. In the RNA-seq data part, line 273-275, authors wrote that they selected 3 genes randomly for validation of their RNA-seq. Instead of randomly selected, why authors did not select top up- or downregulated genes, which are more interesting than randomly selected genes. In relation to this point, data shown in Figure 8A was not delivering a clear meaning.

Comments on the Quality of English Language

Moderate editing is required.

Round 2

Reviewer 3 Report

Comments and Suggestions for Authors

Authors have addressed and significantly improved the manuscript.

Comments on the Quality of English Language

Minor editing is required